# Characterization of Odor-Active Compounds from *Gryllus bimaculatus* Using Gas Chromatography-Mass Spectrometry-Olfactometry

**DOI:** 10.3390/foods12122328

**Published:** 2023-06-09

**Authors:** Hui-Yeong Seong, Eui-Cheol Shin, Youngseung Lee, Misook Kim

**Affiliations:** 1Department of Food Science and Nutrition, Dankook University, Cheonan 31116, Republic of Korea; gmldud7366@gmail.com (H.-Y.S.); youngslee@dankook.ac.kr (Y.L.); 2Department of Green Bio Science/Food Science, Gyeongsang National University, Jinju 52725, Republic of Korea; eshin@gnu.ac.kr

**Keywords:** *Gryllus bimaculatus*, edible insect, volatile compound, gas chromatography-mass spectrometry, gas chromatography-olfactometry

## Abstract

Edible insects have recently attracted attention as an alternative sustainable protein food source. However, consumer aversion remains the major obstacle to successful implementation in the food industry due to their shape and unpleasant odor. Here, we evaluated and compared odor-active compounds from untreated *Gryllus bimaculatus* (UGB), hot-air dried GB at 70 °C for 10 h (AGB), freeze-dried GB (FGB), steam-heated GB at 121 °C and 14.5 psi for 15 min (SGB), and defatted GB by hexane (DFGB). Each sample was analyzed using gas chromatography-mass spectrometry (GC-MS) and gas chromatography-olfactometry (GC-O). The most volatile compounds were detected in UGB, followed by SGB, DFGB, AGB, and FGB by GC-MS analysis. In GC-O analysis, fourteen compounds were identified as cricket or cricket-related odor among twenty identified compounds. Cyclododecane had the strongest cricket-related odor found only in UGB. DFGB received the lowest total scores of intensity for cricket-related odors, while SGB received the highest scores. It seems that defatting could reduce cricket-related odors. This study may provide theoretical information for the GB odors according to the four processing methods.

## 1. Introduction

According to a Food and Agriculture Organization report, the global human population will continue to increase to nine billion by 2050, and global food demand is estimated to increase by approximately 70% [1]. Additionally, the United Nations predicted that by 2100, the global population would exceed 11 billion [2]. With the global human population on the rise and the demand for protein increasing, coupled with the escalating cost of animal protein and mounting environmental concerns, edible insects have garnered significant attention as a highly cost-effective and sustainable alternative to address these challenges. Insects have a high feed conversion ratio, much less water and land requirements, low levels of greenhouse gas emissions, few diseases, and a high percentage of utilizable parts [3,4]. Moreover, edible insects have similar nutritional value to conventional meat [5] and unsaturated fatty acids, which improve blood flow and account for more than 70% of the total fatty acids, making them useful food resources [6].

*Gryllus bimaculatus* (GB), called two-spotted cricket, is a species of the Orthoptera order and the Gryllidae family. They are subtropical insects that are distributed in Africa, Laos, Indonesia, Malaysia, etc. It has been used as a treatment for urinary calculus or diuretic in Europe, and for diarrhea, dysentery, and typhoid in Japan. Additionally, it is cooked and sold as food in the United States, Denmark, and Thailand. GB is an important promising candidate for overcoming the limitations of food supply and protein resources, because it is easy to manage, produces in a sustainable manner, and is rich in protein and minerals such as calcium and zinc [3,7].

Despite these advantages, it is difficult for it to be easily consumed due to its disgusting appearance, peculiar taste, and off-flavor [1,3,4,5,6]. Especially in Western cultures, low entomophagy is observed due to unattractive physical and sensory characteristics. Visual properties, such as appearance, can be improved somewhat by making them invisible in such ways as grinding or hidden in food. The properties related to flavor are still a problem, thus promoting research on flavor improvement. However, GB’s odor profile, which is the basis for flavor research, is yet to be studied.

Edible insects, such as GB, are generally rich in fat and contain lots of unsaturated fatty acids, making them susceptible to lipid oxidation during storage. In addition, while processing and storing edible insects, compositional changes due to inherent microorganisms and enzymes occur. Therefore, an appropriate processing technique, such as drying, defatting, and sterilization, is required for long-term storage and efficient utilization as food materials with quality maintenance. Hot-air drying is inexpensive and convenient and is the most frequently used method to extend the storage period in food manufacturing or processing [8]. The freeze-drying method is more expensive than hot-air drying, but it causes very little quality change and has the advantage of maintaining nutrients and texture [9]. However, dry processing methods have significant effects on sensory or nutritional quality. During the drying process, volatile compounds such as aldehyde, alcohol, and ketone can be formed [2], and several studies have shown that the drying treatment promotes the formation of new volatile substances. These flavor characteristics can have positive or negative impacts on consumers. Removing insect lipids can improve quality and flavor. Choi et al. [5] reported that defatting edible insects such as mealworm larvae, cricket adults, and silkworm pupae by hexane improved protein-specific functions and properties. In a study by Ribeiro et al. [10] involving the sensory evaluation of cereal bars with the whole ground and defatted crickets, defatting improved consumer acceptance of insect products. However, they said that further studies on which components or whole insects were involved in off-flavor were required [5,10]. Additionally, the peculiar flavor generated by microorganisms was reduced by larval sterilization to utilize mealworm larvae as a food ingredient [6].

The purpose of this study is to investigate and compare the effects of different treatments, such as hot-air drying, freeze-drying, steam-heating (sterilization), and defatting, on the odor components of GB. It aims to analyze the volatile compounds present in the odor of GB using gas chromatography-mass spectrometry (GC-MS) and gas chromatography-olfactometry (GC-O) techniques. By examining the changes in the odor profile of GB under different treatments, we sought to determine the impact of these treatments on the sensory characteristics and chemical composition of GB’s odor.

## 2. Materials and Methods

### 2.1. Materials

Forty-day-adult crickets, GB, were obtained in May 2020 from an edible insect farm (Cricket Farm Co., Hwaseong, Republic of Korea). The crickets were used after starvation for 24 h to clear their gastrointestinal tract of any residual food before processing treatment. GB was processed using four treatment methods. Hot-drying was performed at 70 °C for 10 h (AGB). Freeze-drying was performed at −40 °C for 72 h using a freeze dryer (TFD5503, ilshinBioBase Co., Dongducheon, Republic of Korea) (FGB). Steam-heating was performed at 121 °C and 14.5 psi for 15 min using an autoclave (PAC-60, LABHOUSE, Pocheon, Republic of Korea) (SGB). Defatting was performed by removing GB lipids using n-hexane as a solvent, in a solvent-to-sample ratio of 1:20, and stirred for 12 h, then repeated for 48 h (DFGB) at 25 °C [5]. Untreated GB (UGB) was used as a control.

### 2.2. GC-MS Analysis of Volatile Compounds in Five GB Samples

Volatile compounds in the sample set were analyzed using solid-phase microextraction (SPME, Supelco, Bellefonte, PA, USA). After placing a 3 g sample and 1 mL of pentadecane (1 μg/mL) as the internal standard into a 10 mL sized glass vial and sealing the entrance with silicon/Teflon septum, stirred at 100 rpm for 15 min at 60 °C to create an equilibrium state, volatile compounds were adsorbed to fiber (50/30 μm DVB/CAR/PDMS 24 Ga, Supelco, Bellefonte, PA, USA) for 10 min. After that, an inlet fiber of a gas chromatography-mass selective detector (GC-MSD; Agilent 7890A and 5975C, Agilent Technologies, Santa Clara, CA, USA) of 220 °C was inserted, desorbed for 10 min, and then analyzed. MSD was operated in electron impact ionization mode (70 eV), scanning a mass range (m/z) from 30 to 550 amu. The analytical column, HP-5MS (30 m × 0.25 mm, i.d. × 0.25-μm film thickness; Agilent Technologies, Santa Clara, CA, USA), was used, and N_2_ (1 mL/min) was used as a carrier gas. The oven temperature was set to 40 °C for 5 min and then increased by 5 °C/min for it to rise to 200 °C and then maintained for 10 min. The inject temperature was 220 °C and performed in the splitless mode. Individual compounds separated from the total ionization chromatogram of the samples were identified using the mass spectrum library from NIST version 12. The minimum similarity of each peak was over 80%. The peak area of separated volatile compounds was integrated and calculated in proportion to the peak area of pentadecane as the internal standard and expressed as μg/100 g of the sample. RI was calculated according to the following formula and based on a series of n-alkanes (C8–C20):RIx = 100n + 100((tRx − tRn)/(tRn + 1 − tRn))(1)
where RIx is RI of the unknown compound, tRx is the retention time of the unknown compound, tRn is the retention time of the n-alkane, and tRn + 1 is the retention time of the next n-alkane. tRx is between tRn and tRn + 1 (n = number of carbon atoms).

The standard mixture for the retention index was obtained from Sigma–Aldrich company (St. Louis, MO, USA)

### 2.3. GC-Olfactometry Analysis of Odor-Active Compounds in Five GB Samples

GC-O analysis was performed using an olfactory detection port (Gerstel Inc, Linthicum, MD, USA) attached to the Agilent 7890A GC-MSD to detect odors of individual volatile compounds isolated by GC-MS. The sniffing test was performed in four runs by two authors who are conducting this study. The authors were considered as 1 male and 1 female (both are in their 40s), respectively; different genders can respond to different odor descriptions. They took part in the experiment after smelling the sample several times before the experiment and repeating it until they became accustomed to the smell of each sample. Each measurement was conducted simultaneously with the analysis of GC-MS, and the olfactory measurement was conducted for 5–25 min, considering the time when the solvent was eluted within 5 min at the initial stage of analysis and the fatigue of the olfactory sense of the subjects. The intensity and retention time of odor were measured using an intensity sensor divided into four levels (1 = low, 2 = medium, 3 = strong, and 4 = very strong), and the characteristics of volatiles were immediately recorded. All experiments were conducted after the experimenter’s olfactory fatigue was completely recovered. Before the experiment, the experimenters checked the odors identified in all samples and conducted the experiment after discussing the description to express the odors.

## 3. Results and Discussion

### 3.1. Identification of Volatile Compounds by GC-MS Analysis

According to GC-MS analysis, the presence of 142 volatile compounds belonging to fourteen major groups, including forty-three hydrocarbons, nineteen terpenes or terpenoids, sixteen heterocyclic or aromatic compounds, eleven aldehydes, nine alcohols, eight ketones, seven esters, six nitrogen compounds, five acids, four sulfur compounds, three pyrazines, two ethers, one furan, one lactone, and seven others were identified in five kinds of cricket samples (Table 1; Appendix A).

According to the profile of volatile compounds, sixty-six volatile compounds were detected in UGB, such as hydrocarbons, terpenes and terpenoids, heterocyclic or aromatic compounds, aldehydes, alcohols, etc. In SGB, forty-nine volatile compounds were detected, such as terpenes or terpenoids, hydrocarbons, heterocyclic or aromatic compounds, aldehydes, N-containing compounds, etc. In DFGB, forty-nine volatile compounds were detected, such as hydrocarbons, heterocyclic or aromatic compounds, terpenes or terpenoids, aldehydes, alcohols, etc. In AGB, forty-three volatile compounds were detected, such as hydrocarbons, alcohols, heterocyclic or aromatic compounds, terpenes or terpenoids, acids, etc. Forty-three volatile compounds were detected in FGB, such as hydrocarbons, aldehydes, heterocyclic or aromatic compounds, ketones, alcohols, etc. Hydrocarbons were the major group in all samples, except for SGB, but these have a high odor threshold, so it is estimated that they do not have a significant impact on food odor [11]. Terpenes and terpenoids were the major groups in SGB; in addition, they were detected as dominant groups in all kinds of GB. Insects and plants use them for chemical communication; ants produce defense substances related to terpenoids [12]. The biosynthesis of terpenes by terpene synthases in plants has been studied; however, there is little research about the terpene’s mechanism in insects [13]. It is still unclear whether terpenes originated from plants or biosynthesized in many insect species; thus, it seems that further research on this needs to be conducted.

In UGB, sixty-six volatile compounds belonging to eleven groups were detected, and the most volatile compounds were detected among all samples. Heterocyclic or aromatic compounds were detected most in UGB among all samples, with an approximately 16- to 31-fold greater content than in other samples, which means that there were more compounds that contribute to flavoring in UGB than in other samples. The dominant compounds were toluene (324.77 μg pentadecane equivalents/100 g), ethenylbenzene (117.76 μg pentadecane equivalents/100 g), ethylbenzene (97.55 μg pentadecane equivalents/100 g), phytane (78.16 μg pentadecane equivalents/100 g), and benzaldehyde (43.05 μg pentadecane equivalents/100 g). Toluene can be produced or transformed directly from the initial product of sesquiterpenes or monoterpenes by oxidation, dehydration, and other reactions [14]. Benzaldehyde originated from the oxidation of double bonds between carbons in ethenylbenzene. In addition, it is reported to have bitter almond, woody, burned, and metallic odor in Korean anchovy and obscure puffer [15,16]. Because this compound has a low threshold, it is thought to contribute to cricket odor. Of all samples, nonanal was the most detected in UGB. It is the main component of linoleic acid autooxidation and is reported to have oily, plastic, and marine odor [15,16]. Additionally, undecane and dodecane, which were the most detected in UGB among all samples, were reported as alarm substances and defense mechanism compounds in larvae and pupae of the honeybee, respectively [17].

In AGB, forty-three volatile compounds belonging to eleven groups were detected, and the number of compounds was the second least among all samples; the most detected group was the hydrocarbons. Dominant compounds were toluene (20.12 μg pentadecane equivalents/100 g), ethenylbenzene (8.48 μg pentadecane equivalents/100 g), 1,1-dichloro-1-fluoroethane (7.86 μg pentadecane equivalents/100 g), phytane (7.79 μg pentadecane equivalents/100 g), and 2,8-Dimethylundecane (6.76 μg pentadecane equivalents/100 g). Compared with UGB, the contents of toluene and ethenylbenzene were reduced by approximately 16- and 14-fold, respectively.

In FGB, forty-three volatile compounds belonging to ten groups were detected, and the least compounds were detected among all samples; dominant compounds were hexanal (14.48 μg pentadecane equivalents/100 g), benzeneethanol (13.55 μg pentadecane equivalents/100 g), toluene (7.44 μg pentadecane equivalents/100 g), ethenylbenzene (6.31 μg pentadecane equivalents/100 g), and methylthiomethane (3.90 μg pentadecane equivalents/100 g). Compared with UGB, the contents of toluene and ethenylbenzene were reduced by approximately 44- and 19-fold, respectively. Among all samples, the most amount of hexanal was detected in FGB. This compound is derived from the degradation of n-6 polyunsaturated fatty acids oxides and can be used as an indicator of degradation in seafood and meat products. It has also been reported to have a fishy and grassy odor in obscure puffers [15]. Additionally, 2-heptanal and 2-penylfuran, detected only in FGB, were described by raw fish and grassy odor, respectively. Moreover, 1-octen-3-ol, detected only in FGB, has a mushroom-like, earthy, and fungal odor, as well as a fresh fish aroma. This compound is derived from the hydroperoxide degradation of linoleic acid and is considered an off-flavor in porcupine liver [18,19]. Although the least number of volatile compounds were detected in FGB, the proportion of compounds that have negative smells was relatively high. In addition, Kröncke et al. [2] have reported a much more diverse spectrum for lipid oxidation intermediates in freeze-dried larvae.

In SGB, forty-nine volatile compounds belonging to nine groups were detected. Unlike other samples, terpene or terpenoids were the most detected. Terpenes, belonging to the hydrocarbon group, are mostly found in the essential oils of plants, but their flavor is very weak [20]. Dominant compounds were β-phellandrene (79.04 μg pentadecane equivalents/100 g), camphene (21.50 μg pentadecane equivalents/100 g), β-myrcene (13.91 μg pentadecane equivalents/100 g), α-pinene (12.38 μg pentadecane equivalents/100 g), and ethylbenzene (11.80 μg pentadecane equivalents/100 g). Compared with UGB, the main compounds were drastically altered. Toluene and ethenylbenzene, which were the dominant compounds in UGB, decreased by 43- and 11-fold to 7.59 μg pentadecane equivalents/100 g and 10.48 μg pentadecane equivalents/100 g, respectively, and were not the main compounds in SGB.

In DFGB, forty-nine volatile compounds belonging to nine groups were detected. Dominant compounds were hexane (450.84 μg pentadecane equivalents/100 g), ethenylbenzene (11.85 μg pentadecane equivalents/100 g), p-xylene (9.66 μg pentadecane equivalents/100 g), β-terpinene (6.80 μg pentadecane equivalents/100 g), and methyl sulfone (6.19 μg pentadecane equivalents/100 g). High hexane content is due to the treatment method. In the defatting treatment process, hexane was used as a solvent, so it is presumed that hexane residual remained in the sample. However, this compound has an odor threshold of 130 ppm, which has little effect on flavor [21]. The toluene and ethenylbenzene content was significantly reduced by approximately 260- and 10-fold, compared with UGB.

The overall compound contents were the most in UGB (960.3 μg pentadecane equivalents/100 g), followed by DFGB (520.74 μg pentadecane equivalents/100 g), SGB (198.63 μg pentadecane equivalents/100 g), AGB (108.77 μg pentadecane equivalents/100 g), and FGB (64.65 μg pentadecane equivalents/100 g). FGB had the least number of volatile compounds; however, it was not undesirable because of a lot of compounds with negative odor. Even if the volatile compound content in DFGB is the second most among all samples, the dominant compound is hexane (450.84 μg pentadecane equivalents/100 g) used during defatting, which is considered to have no significant effect on flavor. Therefore, it seems that defatting effectively reduced the number of volatile compounds.

Ethenylbenzene, ethylbenzene, and xylene, which were detected in all samples, are known as off-flavors in meat [22], and their contents decreased after all processing treatments. Toluene and p-xylene are derived from environmental pollutants, and these compounds contribute to unpleasant off-flavors [16]. In DFGB, the content of toluene and ethylbenzene was most reduced, and in FGB, the content of ethenylbenzene was most reduced. In conclusion, the total content of volatile compounds the most decreased in FGB, however the content of compounds such as hexanal, 1-octen-3-ol, and 2-pentyl furan, which have a fishy or earthy odor increased; hence it was undesirable. In DFGB, when the hexane content, which is presumed to be the residue of the defatting solvent, was excluded, the total content was the lowest besides the FGB. The different processing methods applied in this study have distinct effects on the volatile composition of GB. These effects can be attributed to mechanisms such as concentration through moisture removal (hot-air drying and freeze-drying), thermal degradation, heat-induced chemical reaction, and facilitation of the release of volatile compounds (hot-air drying and steam heating), and reduction of lipid-associated reaction products by fat removal (defatting).

### 3.2. Odor Description of Major Volatile Compounds Determined by GC-O Analysis

In GC-O analysis, twenty identified compounds were perceived by subjects (Table 2). Among them, thirteen compounds were identified as cricket-related odors.

When calculating the total score of intensity for cricket-related odors, the SGB was seen to be the highest with seven points, and lowest in DFGB with four points. The compound that showed the strongest intensity for the cricket-related odor was cyclododecane, which scored three points. Cyclododecane has a musty odor (GPS Safety Summary) and an unpleasant smell [23]. Butyrolactone, which scored one point in only UGB, is present in meat, fruit, heat-processed food, fermented food, etc. This compound has a low limit of approximately 0.1 ppm, so it has a strong flavor [20] and is reported by a stale and fatty odor [24,25]. Methoxy-phenyl oxime was detected in AGB (2 points), FGB (2 points), and SGB (1 point). It is an N-containing compound having phenyl and methoxy groups. The flavor properties of this compound are not well known. Some studies reported that it is contaminated by SPME fiber. They suggested that it is derived from the glue used for SPME fiber [26]. The compound identified in all samples and most detected in almost all samples by GC-MS analysis was toluene. However, it was detected in only UGB by GC-O analysis. These results are attributed to the threshold of 2.9 ppm toluene [27].

Pentanoic acid, 2,5-dimethyl pyrazine, p-cymene, diisodecyl ether, n-octyl ether, tetramethylpyrazine, and oxalic acid (isobutyl nonyl ester), were identified only in AGB or SGB. These compounds are considered to be degraded or formed from other compounds because both are heat treatment methods. When amino acid and sugar are heated together, an amino-carbonyl reaction occurs. In the final step of the reaction, enaminol is formed by oxidative degradation, known as the Strecker reaction, between the dicarbonyl compound and α-amino acid. It is, therefore, soon cyclized into two molecules, producing pyrazines. Here, 2,5-dimethyl pyrazine and tetramethylpyrazine were described as roasty odors. Pyrazines are known as odor compounds of coffee, peanuts, sesame seeds, and many foods, which are normally heat-treated during processing. Pentanoic acid has an unpleasant odor similar to other low-molecular carboxylic acids [28]. When heat treatment is applied, amino acids and peptides are thermally degraded to cause decarboxylation or deamination, resulting in thermal degradation of proteins that generate aldehydes, hydrocarbons, and N-containing compounds; also, lipids thermal degradation occurs. Lipids are autoxidized to produce free radicals and hydroperoxides, and they form or degrade polymers to produce compounds having rancid odors, such as aldehydes, alcohols, and ketones [20]. Indeed, dried GB contains high protein and lipids at 71% and 16%, respectively, making it easy for thermal degradation to occur [29]. p-Cymene belongs to the terpenes and is known as rancid with a slightly woody oxidized citrus and grassy-kerosene-like odor [30]. Korkmaz et al. [14] reported that this compound may be produced from monoterpene oxidation through isomerization and oxidation.

Overall, when only cricket-related odors were calculated as scores, the score was highest in SGB and AGB, followed by UGB, FGB, and DFGB. Compared with UGB, cricket-related odors increased relatively in AGB and SGB and decreased in FGB and DFGB. The effect of reducing cricket odors was the best in DFGB. Ribeiro et al. [10] reported that sensory evaluation results for cereal bars with defatted crickets were similar to those of the control (without crickets), so they demonstrated that defatting crickets bring about a positive effect. When describing the flavor and smell of the cereal bars with whole ground crickets, words such as moldy, musty, rancid, unpleasant, and old closet were used, and these words are used to describe lipid oxidation. Thus, it can be considered that lipids contribute significantly to the characteristic flavor of crickets. Because of the high risk of lipid oxidation, this suggests that the lipid portion may be associated with crickets’ characteristic flavor.

## 4. Conclusions

The present study aimed to identify the volatile and odor-active compounds in various types of GB that underwent different processing methods, such as hot-air drying, freeze-drying, steam-heating, and defatting. Through GC-MS analysis, we detected a total of 66 volatile compounds in UGB, making it the most abundant, followed by 49 compounds in SGB and DFGB, and 43 compounds in AGB and FGB, which had the lowest number of compounds. Notably, the content of volatile compounds was found to be lowest in FGB. Hydrocarbons were the dominant compounds detected in all samples, with a significant presence of terpenes and terpenoids in SGB. Furthermore, using GC-O analysis, we identified twenty odor-active compounds, of which thirteen had odor descriptions associated with crickets. Cyclododecane received the highest score of three points among the cricket-related odors. Interestingly, the lowest score for the cricket-related odor was observed in DFGB, suggesting that defatting could be an effective method for reducing the peculiar smell of crickets in GB.

In conclusion, our study provides valuable insights into the volatile and odor-active compound composition of GB during different treatment methods. It contributes theoretical knowledge for the development of edible insect-based foods and materials. However, the analysis was limited to a specific set of treatment methods, and there may be other methods or combinations that could yield different results. Based on our outcomes, there are several potential avenues for future studies. Firstly, exploring the sensory attributes of GB using a larger panel group could provide more comprehensive insights into the overall flavor profile. Moreover, investigating the impact of different treatment parameters, such as temperature and duration, on the formation and retention of volatile compounds would enhance our understanding of the underlying mechanisms. Additionally, studying the consumer acceptance and perception of GB products through sensory evaluation could help identify factors influencing the market potential of edible insect-based foods. Finally, evaluating the nutritional composition and potential health benefits of GB in relation to different treatment methods would contribute to the broader field of edible insect-based food industry.

## Figures and Tables

**Table 1 foods-12-02328-t001:** Volatile compounds in five kinds of GB by GC-MS.

Compounds	RI ^(1)^	Contents (μg Pentadecane Equivalents/100 g)					I.D. ^(3)^
		UGB ^(2)^	AGB	FGB	SGB	DFGB	
***Aldehydes***							
Butanal	<800	ND ^(4)^	ND	ND	0.46 ± 0.65	ND	MS
3-Methylbutanal	<800	ND	ND	0.00 ± 0.01	ND	ND	MS
2-Methyl butanal	<800	ND	ND	ND	0.01 ± 0.02	ND	MS
Pentanal	<800	ND	ND	ND	0.02 ± 0.03	ND	MS
Hexanal	820	ND	5.15 ± 4.25	14.48 ± 4.15	ND	1.31 ± 0.04	MS
2-Heptenal	974	ND	ND	0.16 ± 0.23	ND	ND	MS
Benzaldehyde	978	43.05 ± 60.89	0.82 ± 0.69	0.04 ± 0.06	ND	0.81 ± 0.07	MS/RI
Nonanal	1118	11.27 ± 15.94	ND	0.22 ± 0.11	0.63 ± 0.10	1.33 ± 0.02	MS/RI
Ethyl-benzaldehyde	1193	2.03 ± 2.87	ND	ND	ND	ND	MS
Decanal	1218	0.31 ± 0.16	0.43 ± 0.61	0.03 ± 0.05	0.31 ± 0.16	0.36 ± 0.04	MS
2-Butyloct-2-enal	1386	ND	ND	0.09 ± 0.13	ND	ND	MS
***Alcohols***							
1-Octen-3-ol	993	ND	ND	1.25 ± 0.72	ND	ND	MS/RI
2-Butyloctanol	1035	ND	ND	ND	0.08 ± 0.11	ND	MS
3,5-Octadien-2-ol	1060	0.09 ± 0.12	0.60 ± 0.85	0.13 ± 0.18	ND	ND	MS
Benzeneethanol	1129	8.26 ± 11.67	0.81 ± 1.15	13.55 ± 7.72	5.95 ± 0.76	0.57 ± 0.47	MS
2-Hexyldecanol	1141	ND	1.36 ± 1.05	ND	0.33 ± 0.02	0.27 ± 0.01	MS
2-Octyldecan-1-ol	1178	ND	0.05 ± 0.07	ND	ND	0.25 ± 0.03	MS
2-Ethyl-1-hexanol	1243	ND	0.22 ± 0.31	ND	ND	ND	MS
2-Isopropyl-5-methyl-1-hexanol	1318	ND	0.19 ± 0.27	ND	ND	ND	MS
Levomenthol	1188	6.42 ± 9.07	ND	ND	ND	ND	MS
***Ketones***							
2,3-Octanedione	998	ND	ND	0.46 ± 0.18	0.59 ± 0.83	ND	MS
3,6-Dimethyl-4-octanone	988	ND	ND	0.05 ± 0.07	0.37 ± 0.53	ND	MS
6-Methyl-5-hepten-2-one	1001	1.91 ± 2.70	ND	ND	ND	ND	MS
1-Fluoro-2-indanone	1017	ND	ND	ND	ND	0.29 ± 0.07	MS
3-Octen-2-one	1023	ND	ND	0.13 ± 0.19	ND	ND	MS
3-Octen-2-one	1055	ND	2.93 ± 1.88	ND	2.23 ± 0.13	1.61 ± 0.38	MS
Acetophenone	1082	6.67 ± 9.02	1.39 ± 1.97	0.09 ± 0.13	ND	ND	MS
2-Undecanone	1304	0.09 ± 0.12	ND	ND	ND	ND	MS
***Terpenes and terpenoids***							
α-Pinene	952	4.81 ± 6.81	0.18 ± 0.26	ND	12.38 ± 1.18	0.09 ± 0.13	MS/RI
Camphene	966	ND	0.30 ± 0.43	ND	21.50 ± 2.70	0.15 ± 0.21	MS/RI
Sabinene	989	ND	ND	ND	0.59 ± 0.83	ND	MS/RI
2-β-Pinene	989	ND	ND	ND	2.26 ± 2.05	ND	MS
β-Myrcene	1005	0.30 ± 0.43	ND	ND	13.91 ± 1.13	ND	MS
α-Thujene	1020	0.12 ± 0.17	ND	ND	6.44 ± 0.91	0.72 ± 0.06	MS
γ-Terpinene	1023	ND	ND	ND	0.87 ± 0.11	ND	MS/RI
3-Carene	1027	ND	ND	ND	0.48 ± 0.68	ND	MS/RI
α-Terpinene	1033	ND	ND	ND	0.99 ± 0.27	ND	MS/RI
β-Terpinene	1039	1.43 ± 2.02	3.34 ± 4.67	ND	1.24 ± 0.11	6.80 ± 0.21	MS
p-Cymene	1042	ND	ND	ND	1.48 ± 0.30	ND	MS/RI
β-Phellandrene	1046	0.76 ± 1.08	1.80 ± 2.55	2.03 ± 0.42	79.04 ± 5.13	ND	MS/RI
Phytane	1051	78.16 ± 110.40	7.79 ± 8.40	0.10 ± 0.15	ND	2.96 ± 0.71	MS
Squalane	1078	4.12 ± 5.83	ND	0.12 ± 0.11	ND	0.54 ± 0.01	MS
α-Terpinolene	1102	0.14 ± 0.20	ND	ND	1.73 ± 0.11	1.43 ± 0.02	MS
α-Cubebene	1393	10.33 ± 14.62	ND	ND	ND	ND	MS
Caryophyllene	1440	4.99 ± 7.05	ND	ND	ND	ND	MS
β-copaene	1448	2.71 ± 3.84	ND	ND	ND	ND	MS
α-Calacorene	1560	6.72 ± 9.51	ND	ND	ND	ND	MS
***Hydrocarbon***							
Hexane	<800	ND	0.16 ± 0.23	0.32 ± 0.46	ND	450.84 ± 139.35	MS
2,4-Hexadiyne	<800	0.42 ± 0.60	ND	ND	ND	ND	MS
Heptane	<800	ND	ND	0.06 ± 0.09	ND	ND	MS/RI
Octane	821	1.16 ± 1.64	ND	ND	ND	ND	MS
Nonane	916	ND	ND	ND	0.27 ± 0.39	ND	MS
1,3,5,7-Cyclooctatetraene	918	0.72 ± 1.02	ND	ND	ND	ND	MS
2-Methyl-3-ethylheptane	958	ND	ND	ND	0.32 ± 0.45	ND	MS
3-Methylnonane	986	ND	ND	ND	0.81 ± 0.07	ND	MS
2,2,3,4-Tetramethylpentane	1025	ND	0.05 ± 0.07	ND	ND	ND	MS
2,2,3-Trimethyldecane	1026	ND	ND	ND	ND	1.02 ± 1.08	MS
2,2,4,6,6-Pentamethylheptane	1026	0.02 ± 0.03	ND	ND	ND	ND	MS
2,2,9-Trimethyldecane	1039	ND	0.50 ± 0.71	0.06 ± 0.08	ND	0.77 ± 1.08	MS
2,6,8-Trimethyl-decane	1055	ND	ND	1.76 ± 2.50	ND	ND	MS
2,6-Dimethyloctane	1069	11.91 ± 16.85	ND	ND	ND	2.70 ± 0.09	MS
4,6-Dimethylundecane	1071	ND	0.82 ± 1.15	0.42 ± 0.59	ND	0.63 ± 0.02	MS
2-Methyl-decane	1073	0.59 ± 0.02	0.14 ± 0.20	0.13 ± 0.18	ND	0.59 ± 0.02	MS
3,7-Dimethylnonane	1084	ND	ND	ND	0.79 ± 0.06	ND	MS
3-Ethyl-3-methylheptadecane	1087	ND	ND	ND	ND	0.19 ± 0.00	MS
5-Methyl-octadecane	1087	ND	0.54 ± 0.59	ND	ND	ND	MS
2,8-Dimethylundecane	1090	8.29 ± 11.72	6.76 ± 7.50	0.16 ± 0.22	ND	2.88 ± 0.16	MS
3-Ethyl-3-methylheptane	1097	ND	1.10 ± 1.27	ND	ND	0.33 ± 0.00	MS
4-Methylundecane	1107	ND	0.16 ± 0.22	0.03 ± 0.04	ND	0.55 ± 0.02	MS
1-Phenyl-1-butene	1109	8.95 ± 12.49	ND	ND	ND	ND	MS
Undecane	1112	10.41 ± 14.72	2.78 ± 3.93	ND	1.42 ± 0.06	1.42 ± 0.05	MS
2,8-Dimethylundecane	1147	ND	0.37 ± 0.53	ND	0.06 ± 0.09	ND	MS
2-Methyl-1-tetradecene	1180	ND	ND	ND	ND	0.09 ± 0.01	MS
2-Ethyl-decane	1182	ND	ND	ND	0.05 ± 0.07	0.15 ± 0.06	MS
2,4-Dimethyl-1-heptene	1196	ND	ND	ND	ND	0.14 ± 0.01	MS
Cyclododecane	1202	6.15 ± 8.69	ND	ND	ND	ND	MS
Trans-2-nonadecene	1208	0.13 ± 0.19	ND	ND	ND	0.06 ± 0.01	MS
Dodecane	1211	10.78 ± 15.24	0.78 ± 1.10	0.08 ± 0.05	0.37 ± 0.37	0.29 ± 0.10	MS
5-Dodecene	1255	ND	ND	ND	ND	0.11 ± 0.01	MS
Tridecane	1310	6.82 ± 8.99	0.25 ± 0.35	0.02 ± 0.03	0.14 ± 0.03	0.25 ± 0.13	MS
11-Decyltetracosane	1317	ND	ND	ND	ND	0.07 ± 0.10	MS
3-Tetradecene	1356	1.85 ± 2.62	ND	ND	ND	ND	MS
Tetradecane	1408	8.44 ± 11.89	ND	0.42 ± 0.29	ND	0.08 ± 0.11	MS
3-Eicosene	1453	3.11 ± 4.39	ND	ND	ND	ND	MS
trans-7-pentadecene	1491	0.04 ± 0.05	ND	ND	ND	ND	MS
3-Ethyl-tetracosane	1504	0.02 ± 0.03	ND	ND	ND	ND	MS
Hexadecane	1605	8.50 ± 12.02	ND	0.28 ± 0.38	ND	ND	MS
Heptadecane	>1700	14.80 ± 20.93	ND	ND	ND	ND	MS
Octadecane	>1700	9.97 ± 14.10	ND	ND	ND	ND	MS
Nonadecane	>1700	12.88 ± 18.22	ND	ND	ND	ND	MS
***Nitrogen-containing compounds***							
Trimethylamine	<800	ND	ND	ND	0.05 ± 0.07	ND	MS
2-Butanone oxime	809	ND	0.17 ± 0.24	ND	ND	ND	MS
Ethyl methyl ketone oxime	813	ND	ND	ND	2.15 ± 1.05	ND	MS/RI
Penicillamine	836	ND	ND	ND	ND	0.17 ± 0.01	MS
Methoxy-phenyl-oxime	921	0.40 ± 0.56	1.12 ± 1.59	0.34 ± 0.19	0.57 ± 0.80	ND	MS
1,2-Benzenediamine	933	ND	ND	ND	0.04 ± 0.05	ND	MS
Subtotal content		0.4	1.29	0.34	2.81	0.17	
***Acids***							
Pentanoic acid	867	ND	0.15 ± 0.21	ND	ND	ND	MS/RI
2-Methylbutanoic acid	875	ND	0.58 ± 0.83	ND	ND	ND	MS
Sobutylacetic acid	967	ND	0.14 ± 0.20	ND	ND	ND	MS
Nonahexacontanoic acid	>1700	4.87 ± 6.89	ND	ND	ND	ND	MS
Docosanoic acid	>1700	14.89 ± 21.06	ND	ND	ND	ND	MS
***Esters***							
Oxalic acid, isobutyl nonyl ester	1134	ND	0.96 ± 1.36	ND	ND	ND	MS
Caprylyl acetate	1165	2.72 ± 3.85	ND	ND	ND	ND	MS
Acetic acid, phenethyl ester	1271	ND	ND	0.06 ± 0.09	ND	ND	MS
Ethyl salicylate	1285	ND	ND	ND	ND	ND	MS
Pentafluoropropionic acid, tetradecyl ester	1293	ND	ND	ND	ND	0.08 ± 0.01	MS
Diisopropyl adipate	1465	0.03 ± 0.04	ND	0.10 ± 0.14	ND	ND	MS
Triacontyl trifluoroacetate	>1700	11.17 ± 15.79	ND	ND	ND	ND	MS
***Heterocyclic or aromatic compounds***							
Benzene	<800	1.24 ± 1.76	ND	ND	ND	3.09 ± 0.34	MS
2-Methylpiperazine	<800	ND	ND	0.06 ± 0.04	ND	ND	MS
Toluene	<800	324.77 ± 422.17	20.12 ± 7.67	7.44 ± 0.44	7.59 ± 0.03	1.25 ± 0.07	MS
5-Methyl-2-phenylindole	839	ND	ND	ND	ND	0.42 ± 0.59	MS
Ethylbenzene	881	97.55 ± 134.67	3.30 ± 0.98	2.30 ± 0.62	11.80 ± 3.55	2.13 ± 0.03	MS
p-Xylene	888	5.85 ± 6.79	2.84 ± 1.75	1.73 ± 0.15	0.41 ± 0.57	9.66 ± 1.44	MS
m-Xylol	889	0.55 ± 0.78	1.13 ± 1.60	0.66 ± 0.31	3.22 ± 4.55	ND	MS
Ethenylbenzene	909	117.76 ± 151.04	8.48 ± 3.71	6.31 ± 2.52	10.48 ± 3.46	11.85 ± 0.70	MS
1,3-Dimethylbezene	983	ND	ND	ND	0.14 ± 0.20	ND	MS
Phenol	996	12.81 ± 18.11	ND	ND	ND	ND	MS
Benzenol	997	0.08 ± 0.12	ND	ND	ND	ND	MS
1,2,3-Trimethylbenzene	1029	1.15 ± 1.62	ND	ND	ND	ND	MS
Naphthalene	1198	11.96 ± 16.79	ND	ND	ND	0.36 ± 0.04	MS
7-Butyldocosane	1258	ND	ND	ND	0.07 ± 0.01	ND	MS
o-Methylbiphenyl-diphenylmethane	1415	9.51 ± 13.44	ND	ND	ND	ND	MS
Butylated Hydroxytoluene	1527	ND	ND	ND	ND	0.03 ± 0.05	MS
***Furans***							
2-Pentyl furan	1005	ND	ND	1.35 ± 0.79	ND	ND	MS
***Pyrazines***							
2,5-Dimethyl pyrazine	930	ND	ND	ND	1.12 ± 0.33	ND	MS
2-Ethyl-3,6-dimethylpyrazine	1093	ND	ND	ND	0.88 ± 0.12	ND	MS
Tetramethylpyrazine	1098	ND	1.24 ± 1.76	ND	1.07 ± 0.09	ND	MS
***Sulfur-containing compounds***							
Methylthiomethane	<800	ND	ND	3.90 ± 5.52	ND	ND	MS
Dimethyl sulfide	<800	ND	ND	2.30 ± 0.18	ND	ND	MS
Methyl sulfone	941	ND	0.43 ± 0.27	ND	ND	6.19 ± 8.73	MS/RI
1-Octadecanesulphonyl chloride	1467	4.89 ± 6.91	ND	ND	ND	ND	MS
***Lactones***							
Butyrolactone	940	7.67 ± 10.84	ND	ND	ND	ND	MS
***Ethers***							
Diisodecyl ether	1067	ND	ND	ND	0.19 ± 0.07	ND	MS
n-Octyl ether	1069	ND	ND	ND	0.07 ± 0.10	ND	MS
**Etc.**							
1,1-Dichloro-1-fluoroethane	<800	ND	7.86 ± 11.11	ND	ND	ND	MS
1,1-Difluorodecane	820	0.01 ± 0.01	ND	ND	ND	ND	MS
Tetrachloroethylene	837	1.26 ± 1.79	ND	ND	ND	ND	MS
Pentalin	991	8.49 ± 12.01	ND	1.25 ± 0.72	ND	ND	MS
1-Bromopentadecane	1121	ND	ND	0.11 ± 0.15	ND	ND	MS
3-Heptafluorobutyroxytridecane	1299	ND	ND	ND	ND	0.03 ± 0.01	MS
1,22-Dibromodocosane	1380	ND	ND	ND	ND	0.05 ± 0.01	MS
**Total content**		**960.28**	**90.29**	**64.58**	**197.97**	**517.96**	

^(1)^ RI: retention index. ^(2)^ UGB: untreated GB, AGB: air-dried GB, FGB: freeze-dried GB, SGB: steam-heated GB, DFGB: defatted GB. ^(3)^ I.D: identification; MS: compound identified by comparison with the NIST12 mass spectral database; RI: compound confirmed by retention index. ^(4)^ ND: not detected.

**Table 2 foods-12-02328-t002:** Major compounds and odor descriptions in five kinds of GB by GC-O.

No.	Major Compounds	RI ^(1)^	Odor Description	Odor Intensity ^(2)^					I.D. ^(4)^
				UGB ^(3)^	AGB	FGB	SGB	DFGB	
1	Toluene	<800	roasty, aromatic	1	ND ^(5)^	ND	ND	ND	MS/RI
2	Hexanal	820	grass	ND	ND	1	ND	ND	MS/RI
3	Pentanoic acid	867	salt smell	ND	3	ND	ND	ND	MS/RI
4	Methoxy-phenyl-oxime	921	cricket	ND	2	2	1	ND	MS
5	2,5-Dimethyl pyrazine	930	roasty	ND	ND	ND	1	ND	MS/RI
6	Butyrolactone	940	cricket	1	ND	ND	ND	ND	MS
7	Methyl sulfone	941	hot dried cricket, de-fatted cricket, roasty	ND	1	ND	ND	1	MS
8	α-Pinene	952	grass	ND	ND	ND	1	ND	MS/RI
9	1-Octen-3-ol	993	bitter smell	ND	ND	1	ND	ND	MS
10	p-Cymene	1042	cricket	ND	ND	ND	1	ND	MS/RI
11	3-Octen-2-one	1055	cricket	ND	ND	ND	1	ND	MS
12	Diisodecyl ether	1067	cricket	ND	ND	ND	1	ND	MS
13	n-Octyl ether	1069	autoclaved cricket, roasty	ND	ND	ND	2	ND	MS
14	Tetramethylpyrazine	1098	roasty, Barley	ND	1	ND	1	ND	MS
15	Benzeneethanol	1129	cricket	2	2	2	1	1	MS
16	Oxalic acid, isobutyl nonyl ester	1134	cricket	ND	2	ND	ND	ND	MS
17	2-Hexyldecanol	1141	defatted cricket, roasty	ND	ND	ND	ND	1	MS
18	2-Octyldecan-1-ol	1178	cricket	ND	ND	ND	ND	1	MS
19	Cyclododecane	1202	cricket	3	ND	ND	ND	ND	MS
20	Acetic acid, phenethyl ester	1271	freeze-dried cricket	ND	ND	1	ND	ND	MS

^(1)^ RI: retention index. ^(2)^ Odor intensity was described as the relative intensity (1 to 4) of each volatile compound coming out of the ODP port by experimenters. ^(3)^ UGB: untreated GB, AGB: air-dried GB, FGB: freeze-dried GB, SGB: steam-heated GB, DFGB: defatted GB. ^(4)^ I.D: identification; MS: compound identified by comparison with the NIST12 mass spectral database; RI: compound confirmed by retention index. ^(5)^ ND: not detected.

## Data Availability

The data presented in this study are available on request from the corresponding author.

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
