# Peer review of "Characterization of Odor-Active Compounds from Gryllus bimaculatus Using Gas Chromatography-Mass Spectrometry-Olfactometry"

_foods, 2023, doi:10.3390/foods12122328_

Round 1

Reviewer 1 Report

The study on is edible insect aroma is quite interesting, but the research methodology is too classic, and there are some parts of the report should be clarified by Authors:

- Abstract: the mentioning of important compounds should be more appealing with odor description and their impacts to the overall aroma changes due to different processing methods.

1. Introduction:
1) Lines 52-55: Sentences need references. Also please add the information from the literatures regarding odor description for the off-flavor of edible insect.
2) Review on processing methods used for the study should be well described.
3) Review on analytical method used for the study, including what kind of GC-O technique has been used should also be well described.
4) Originality of the study should be clearly defined.

2. Materials and Methods
1) Lines 82-83: describe the developmental stage of insect and sample collection date.
2) Lines 103-104: state the minimum similarity index for MS identification should be stated.
3) Lines 104:106: state the concentration and added volume of internal standard.
4) Lines 106-107: describe the method for calculating the RI, including the range of alkane.
5) Lines 111-112: state the panelists' age and sex, and the requirement to become the subjects in the GC-O analysis.
6) Lines 117-120: delete the following: 3. Results and so on

3. Results and Discussion
- Table 1:  state the statistical differences of each compound of UGB and other processing methods, and then discuss it accordingly.
- Table 2:
1) State the meaning of odor intensity at the table footnote.
2) Is there any variation(s) on perceived odor intensities for all compounds between the two subjects? If yes, show it.
- Tables 1 and 2:
1) RT is not necessary to be shown.
2) Why there are so many compounds without RI confirmation?
3) On the other hand, how Authors could conclude RI identifications for butanal, 3-methylbutanal, 2-methylbutanal, and pentanal where their RIs were less than <800, and thus could not be confirmed.
Figure 1: state the statistical differences of total compounds of UGB and other processing methods, and then discuss it accordingly.
- Discussion on the impacts processing methods to the volatile composition should be well elaborated. Why would these processes alter some certain of compounds? Discuss the possible mechanisms for each processing method!

4. (not 5) Conclusions
Write both importance and limitations of the study.
Also write the possible future studies based on your current outcomes. 

Authors should seek helps from professional proofreader when revising their manuscript.

Author Response

We appreciate your description with valuable comments about our manuscript. We worked to the best of our abilities to revise the issues as you suggested. The modification was red colored

Reviewer 2 Report

The manuscript fits with the scope of the journal, and represents an interesting analysis of volatiles from a novel source of food.

Line 99: Write up supplier/ manufacturer of the column.

Line 102: ‘The injector/desorption temperature’ rather than ‘The inlet temperature’.

Paragraph 2.2:

-Describe the MS-settings of the instrument (method).

-How was the chromatogram obtained from which peak areas were measured? TIC-chromatograms? Explain.

-How and why were retention indexes used? How were they calculated?

-Give an example of a chromatogram with ‘cricked’-compounds.

Table 1: Rather:

-Contents (µg pentadecane equivalents/ 100g)

-What are significant numbers? 2-digits really? Check up!

Language and style should be improved. E.g.:

Line 15: Evaluate rather than evaluated.

Line 20: Remove ‘..13 compounds, such as…’

Line 36: Remove ‘..as problems, such as the,..’

Line 99: Rather: The analytical column…

Line 127: One furan

Line 127:  ..were identified..

Line 259: Remove ‘such as’ and rewrite.

Line 261: Remove ‘or both’

Author Response

(The authors gave the same response as above.)

Reviewer 3 Report

“Characterization of Odor-Active Compounds from Gryllus bimaculatus using Gas Chromatography-Mass Spectrometry- 3 Olfactometry” (manuscript ID 2430653) is the study about the odor compounds in Gryllus bimaculatus during various treatments and how these treatments influence the contents of these compounds. The results of this study can useful for the development of insects as an alternative protein source. However, following points are recommended to revise, along with thorough revision of English language:

Title needs to be revised to make it comprehensive by including “various treatment” or similar terms?

Abstract (Line 15-20): The description of the methods followed along with the analytical tools used are mix-matched and the readers may get confused. Please revise this sentence by creating two sentences for the processing methods and analytical methods.

Line 35-37: “With this increase in the human population, as problems, such as the increasing demand for protein, the rising cost of animal protein, and growing concerns about environmental issues emerge,…..” Please revise this section to make the meaning clear.

Line 62: “…… freeze-drying are inexpensive…..” – probably, freeze drying is considered as an expensive method?

Line 76-79: The aim of the study needs to be clearly written here?

Line 83: “The crickets were used after starvation for 24 h….” Please specify the reason behind the 24 h starvation before use.

Degree centigrade should be “oC” – throughout the manuscript.

Section 2.1: Freeze-drying and Defatting procedure should be detailed with the parameters (temperature, time etc.) selected for the process.

Statistical analysis section is missing in Materials and Methods.

Table 1 and Table 2: MS and MS/RI – please add the full forms of them in footnotes.

What is the difference between Table 1 and Fig. 1? As the detail list of compounds along with their groupings have been illustrated in Table 1, the presentation of Fig. 1 seems unnecessary? What do you think?

Reference listing needs thorough revision in terms of Title UPPER case initials and Italic scientific names.

Thorough revision of English language is recommended!

Author Response

We appreciate your description with valuable comments about our manuscript. We worked to the best of our abilities to revise the issues as you suggested. The modification was red colored.

Reviewer 4 Report

Dear Authors,

Despite the manuscript resulted well written and the methodological approach is correct, there are some issues to improve, clarify, and justify.

-      Line 94. How much was the volume of the vial?

-      Line 106. Better specify how the retention indices were calculated.

-      Line 117-119. Remove the sentence.

Author Response

(The authors gave the same response as above.)

Reviewer 5 Report

I thought the paper was interesting looking at the VOC profile of GB that have undergone four different processing treatments.

1. I would recommended adding the abbreviations of your treatments to the body of the text as well. You have it listed in the abstract and tables, but I feel it needs to be added to the methods as well.

2. How were the panelists trained? Do you have an IRB or human studies documentation to show approval for this study? Why were these panelists selected over others? 

Author Response

(The authors gave the same response as above.)
